# Comparative Study of Polymer-Grafted BaTiO_3_ Nanoparticles Synthesized Using Normal ATRP as Well as ATRP and ARGET-ATRP with Sacrificial Initiator with a Focus on Controlling the Polymer Graft Density and Molecular Weight

**DOI:** 10.3390/molecules28114444

**Published:** 2023-05-30

**Authors:** Ikeoluwa E. Apata, Bhausaheb V. Tawade, Steven P. Cummings, Nihar Pradhan, Alamgir Karim, Dharmaraj Raghavan

**Affiliations:** 1Department of Chemistry, Howard University, Washington, DC 20059, USA; apatabett@gmail.com (I.E.A.); bvtawade@gmail.com (B.V.T.); steven.cummings@howard.edu (S.P.C.); 2Department of Chemistry, Physics and Atmospheric Science, Jackson State University, Jackson, MS 39217, USA; nihar.r.pradhan@jsums.edu; 3Department of Chemical and Biomolecular Engineering, University of Houston, Houston, TX 77204, USA; akarim3@central.uh.edu

**Keywords:** PS-g-BaTiO_3_ nanoparticles, PMMA-g-BaTiO_3_ nanoparticles, ATRP, sacrificial initiator, ARGET

## Abstract

Structurally well-defined polymer-grafted nanoparticle hybrids are highly sought after for a variety of applications, such as antifouling, mechanical reinforcement, separations, and sensing. Herein, we report the synthesis of poly(methyl methacrylate) grafted- and poly(styrene) grafted-BaTiO_3_ nanoparticles using activator regeneration via electron transfer (ARGET ATRP) with a sacrificial initiator, atom transfer radical polymerization (normal ATRP), and ATRP with sacrificial initiator, to understand the role of the polymerization procedure in influencing the structure of nanoparticle hybrids. Irrespective of the polymerization procedure adopted for the synthesis of nanoparticle hybrids, we noticed PS grafted on the nanoparticles showed moderation in molecular weight and graft density (ranging from 30,400 to 83,900 g/mol and 0.122 to 0.067 chain/nm^2^) compared to PMMA-grafted nanoparticles (ranging from 44,620 to 230,000 g/mol and 0.071 to 0.015 chain/nm^2^). Reducing the polymerization time during ATRP has a significant impact on the molecular weight of polymer brushes grafted on the nanoparticles. PMMA-grafted nanoparticles synthesized using ATRP had lower graft density and considerably higher molecular weight compared to PS-grafted nanoparticles. However, the addition of a sacrificial initiator during ATRP resulted in moderation of the molecular weight and graft density of PMMA-grafted nanoparticles. The use of a sacrificial initiator along with ARGET offered the best control in achieving lower molecular weight and narrow dispersity for both PS (37,870 g/mol and PDI of 1.259) and PMMA (44,620 g/mol and PDI of 1.263) nanoparticle hybrid systems.

## 1. Introduction

Polymer nanocomposites have been the focus of considerable research over the past several decades due to their ability to produce materials with a wide range of advanced properties [1,2,3,4]. BaTiO_3_ nanoparticles are of particular interest among dielectric materials due to their superior dielectric permittivity, piezoelectric and ferroelectric properties, flexibility in adjusting their shape, and widespread use in energy storage, electronics, and device applications [5,6]. As a result, numerous methodologies for preparing BaTiO_3_-based nanocomposites have been devised in order to increase the compatibility of the nanoparticles with the polymer matrix, the spatial dispersion of the nanoparticles in the matrix, and address the permittivity contrast between polymer and nanoparticles [7]. An essential building block in formulating single-component or multicomponent polymer nanocomposites is the use of polymer-grafted nanoparticles hybrids [8,9,10,11]. Two main approaches have been used to synthesize polymer-grafted nanoparticle hybrids. The “*grafting to*” approach involves the attachment of end-functionalized polymer chains on the surface of nanoparticles via esterification, silylation, or click reactions, such as thiol-ene or alkyne-azide cycloaddition, to surface-activated nanoparticles [8]. An advantage of the “grafting to” approach is a monodisperse polymer on a nanoparticle surface can be achieved, allowing it to serve as an excellent model system. However, the bulkiness of long preformed polymer chains and steric hindrance limits the extent of grafting on the nanoparticle surface. More recently, new methods claim higher grafting densities can be achieved, even with the “grafting to” method [12,13].

The “*grafting-from*”, also called surface-initiated control radical polymerization (*SI-CRP*), approach is based on growing polymer chains directly from the surface of nanoparticles functionalized with suitable initiator/CTA functionalities. The advantage of the “grafting from” method is the potential to achieve high grafting densities due to the ease of attaching a high number of monomers on the nanoparticle surface. However, the “grafting from” approach can be challenging because of the complex workup procedure, as well as particle aggregation. Many of these challenges have been addressed through the pioneering work of Matyjaszewski, [14] Mueller, [15] Benicewicz, [16] Takahara, [17] Hawker, [18], and coworkers. For example, good control over molecular weight distribution (MWD) in ATRP needs a certain deactivator concentration together with fast initiation to achieve uniform chain growth [19,20]. Typically, in conventional ATRP, a sufficiently high Cu catalyst concentration is needed to reach high monomer conversion [20,21,22]. However, the presence of a high amount of transition metal complexes may complicate polymer-grafted nanoparticle purification [22,23]. Therefore, minimal Cu(I) catalyst concentrations in ATRP are desirable but the trade-off is the broadening of MWD in grafted chains [24]. 

ARGET ATRP, indeed, addresses the purification of polymer-grafted nanoparticles, a problem arising due to the use of excessive copper catalysts during ATRP, by employing a lower copper concentration. The addition of sannous octoate (Sn(Oct)_2_) as a reducing agent allows for continuous regeneration of Cu(I) (ATRP activator) from Cu(II) (ATRP deactivator), thereby lowering the total copper concentration needed to perform ATRP [24]. By adopting this method, Matyjaszewski et al. [25] demonstrated the synthesis of polystyrene (PS) grafts on SiO_2_ nanoparticles (15 nm), with a low polydispersity of Đ = 1.17, using only 10 parts-per-million concentrations of copper (II) catalyst/L in relation to monomer [25]. However, it must be noted that the amount of Cu(II)Br_2_ catalyst needed for grafting a polymer onto a nanoparticle may ultimately depend on the amount, size, and type of nanoparticles, and the amount of solvent and monomer used. By adopting a similar procedure in this study, we kept the concentration ratio of initiator-modified nanoparticles to CuBr_2_ at 2x:1 similar to that reported by Matyjaszewski et al. [25] for 200 ppm nanoparticles to 100 ppm of Cu(II)Br_2_.

In addition to the use of a Cu(II) pre-catalyst along with a reducing agent to synthesize grafted nanoparticles with less broadened MWD, a sacrificial initiator has also been used in the reaction medium [25,26,27,28]. The sacrificial initiator serves two purposes: one is to increase control over polymerization on the surface and the second is simply for analysis. Diffusion-controlled deactivation was enhanced when a sacrificial initiator was added to the polymerization mixture, playing a similar role to the well-known Trommsdorff effect, which brings about an increase in the polymerization rate for free-radical polymerization at high conversions [25,26,27,28]

Although a significant amount of research has been conducted on the grafting of polymers on SiO_2_ nanoparticles using normal ATRP as well as ARGET ATRP [8,25,29], there has been limited work conducted on the synthesis of polymer-grafted BaTiO_3_ nanoparticles using normal ATRP and ARGET ATRP [25,30]. More importantly, the use of a sacrificial initiator to control polymerization on the surface of BaTiO_3_ nanoparticles using ATRP as well as ARGET has not been well studied outside of a few literature reports [31,32,33,34,35,36,37]. Thus, the primary objective of this study was to compare the graft density, dispersity, and molecular weight of polymer-grafted BaTiO_3_ nanoparticles synthesized using ATRP and ARGET-ATRP techniques, with and without a sacrificial initiator. Our grafting study was not only limited to reactive methyl methacrylate (MMA) monomer but also extended to less reactive styrene monomers in the synthesis of PMMA-g-BaTiO_3_ and PS-g- BaTiO_3_, respectively. These grafted nanoparticles are widely used in several applications and serve as useful models for experimental studies. 

## 2. Results and Discussion

For the synthesis of core–shell-structured PMMA-g-BaTiO_3_ and PS-g-BaTiO_3_ nanoparticles using ATRP and ARGET ATRP methods, the commercially available BaTiO_3_ nanoparticles were subjected to a series of reactions, such as hydroxylation, silanization, and initiator attachment, to prepare the nanoparticle surface for polymer chain growth (Figure 1 and Figure 2). Figure 1A shows the IR for the steps performed to obtain initiator-grafted BaTiO_3_ nanoparticles. The successful modification of nanoparticles was established via the appearance of a broad –OH absorption stretching band around 3400–3600 cm^−1^ upon the hydroxylation of BaTiO_3_ nanoparticles, an alkyl stretching band at 2929 cm^−1^ due to the aliphatic –C-H and at 1566 cm^−1^ due to the N-H stretch upon silylation of the BaTiO_3_ nanoparticles with APTMS; a carbonyl stretching band at 1650 cm^−1^ due to –CONH- (amide) upon initiator functionalization of BaTiO_3_ nanoparticles with BIB. These observations were further substantiated by performing TGA analysis of the modified nanoparticles to successfully demonstrate the surface activation, functionalization, and modification of the BaTiO_3_ nanoparticles with OH, APTMS, and BIB functional groups, respectively. As expected a gradual increase in weight loss in the order of BaTiO_3_ < BaTiO_3_-OH < BaTiO_3_-APTMS < BaTiO_3_-BIB was observed in the thermogram, which is consistent with the molar mass of functionality attached to the nanoparticles [35]. With the increase in grafted organic component onto nanoparticles, a more substantial drop in char yield was noticed.

The grafting reaction of styrene and MMA on the surface BaTiO_3_-BIB was carried out via ATRP in a “grafting from” approach by adopting the procedure previously reported by Xie et al. [35]. The FT-IR spectra (Figure 2A) of PMMA-g-BaTiO_3_ show characteristic absorption bands at 2939 cm^−1^ (C–H groups), 1726 cm^−1^ (–C=O stretching), 1247 cm^−1^ (C-O stretching), and 1153 cm^−1^ (C-O-C stretching) **[38]**. Likewise, the FT-IR spectra (Figure 2B) for the PS-g-BaTiO_3_ show a strong band at 3030 cm^−1^ (-CH aromatic stretching group), 1601 cm^−1^, 1490 cm^−1^, and 751 cm^−1^ (C=C aromatic group), consistent with the previously described observation for polystyrene grafted onto nanoparticles [39]. The intensity of the carbonyl peak of PMMA in PMMA-g-BaTiO_3_ nanoparticles and the aromatic peak of PS in PS-g-BaTiO_3_ nanoparticles varied with reaction time and procedure adopted. For example, a significant difference in the intensity of the FT-IR peaks (Figure 2A,B) of 24 h ATRP and the 12 h ARGET with a sacrificial initiator of PS and PMMA-grafted BaTiO_3_ nanoparticles was noticed. More details about the influence of reaction time and condition on the characteristic of the polymer grafted on the nanoparticles will be discussed later.

To validate the successful grafting of the polymer on the surface of the nanoparticles, PS- and PMMA-grafted BaTiO_3_ nanoparticles were subjected to a cleavage experiment. The cleaved polymer was characterized via ^1^H-NMR and GPC analysis. The ^1^H-NMR spectrum of the cleaved polymer from PS-g-BaTiO_3_ nanoparticles (Figure 3) shows all the characteristic proton peaks for polystyrene. The aromatic protons (**c**, δ 6.56–7.31 ppm, =C−H), methylene (**b**, 0.82–1.24 ppm, −C−H), and the methine peak (**a**, 1.55–2.16, C−H) were observed in the NMR spectrum. Similarly, the ^1^H-NMR spectrum for cleaved PMMA from PMMA-grafted BaTiO_3_ shows methoxy methylene and methyl peaks, confirming the successful grafting of the PMMA on the polymer-grafted BaTiO_3_ nanoparticle. The NMR spectrum for PMMA-g-BaTiO_3_ was reported in our previous paper [7].

Generally, a sacrificial initiator is added to the polymerization mixture and the supernatant is separated from the nanoparticles and precipitated to obtain free polymer chains that are grown in solution, as analyzed using GPC [40]. The major drawbacks to using this approach are that it neglects possible surface confinement and diffusion of the reactive components to and from the propagating radical ends and can result in different molecular weights and dispersity of surface-bound polymer chains compared to a free polymer in solution [41]. In this study, accurate molecular weight and dispersity of the grafted polymer were obtained by cleaving the polymer directly from the nanoparticle. A comparison the molecular weight and dispersity of the free polymer in solution was also determined to validate the aforementioned observation. For example, we observed that the molecular weight and dispersity of the free polymer chains (12 h ATRP with a sacrificial initiator was 16,990 g/mol and 1.213) were lower than that of the polymer cleaved (12 h ATRP with a sacrificial initiator was 30,400 g/mol and 1.522) from the surface of the nanoparticle. Consequently, in our subsequent studies, we determined the molecular weight of the cleaved polymer from the nanoparticles instead of analyzing the molecular weight of the polymer from solution. 

Additionally, we performed transmission electron microscopy (TEM) measurements of the PS- and PMMA-grafted BaTiO_3_ nanoparticles to observe the morphology and the dispersion of the homopolymer-grafted nanoparticles. Figure 4 shows representative TEM images of the polymer-grafted BaTiO_3._nanoparticles. The TEM image shows that the nanoparticles are nearly spherical or elliptical or cubic shaped. As expected, the TEM micrograph provides direct visual and clear evidence of two different regions in the polymer-grafted nanoparticles. We notice that the corona of nanoparticles has a different contrast and lighter texture (corresponding to less electron density polymeric material) than the center. It can be observed that the nanoparticles are homogeneously encapsulated by a layer of polymer, and the polymer-grafted nanoparticles are well dispersed. This shows that the grafting-from approach used in this study is indeed successful. As shown in Figure 5, we observed the polymer shell thickness to vary depending on the time and procedure used for the synthesis of polymer-grafted nanoparticles. The shell thickness of the 24 h PMMA-g-BaTiO_3_ and PS-g-BaTiO_3_ (Figure 5A,B) is approximately 17 nm and 10 nm, respectively, which is consistent with the higher molecular weight obtained for the 24 h PMMA-g-BaTiO_3,_ as compared to PS-g-BaTiO_3._ Similar results were observed for the 12 h ARGET ATRP reaction with a sacrificial initiator. The shell thickness of PS-g-BaTiO_3_ and PMMA-g-BaTiO_3_ (Figure 5C,D) was approximately 7 nm and 5 nm, respectively. This observation is consistent with the lower molecular weight and lower char yield obtained for PS-g-BaTiO_3_ and PMMA-g-BaTiO_3_ nanoparticles after 12 h polymerization. These results further establish that the polymer was successfully grafted onto the surface of the BIB-modified BaTiO_3_ nanoparticles via the various procedures adopted.

PXRD analysis was carried out to determine what, if any, effects grafting had on the nanoparticle. Figure 6 presents the normalized XRD patterns of commercial BaTiO_3_ and polymer-grafted BaTiO_3_ nanoparticles. The PXRD patterns of the purchased and polymer-grafted nanoparticles exhibited characteristic crystallographic peaks at 22.2°, 31.5°, 38.9°, 45.2°, 50.9°, 56.2°, 65.9°, 70.4°, 74.9°, 79.2°, 83.5°, and 87.7°, as described in the figure. Using Bragg’s law, the Miller indices were determined to be (100), (110), (111), (200), (210), (211), (220), (300), (310), (311), (222), and (320) for a face-centered cubic structure, in agreement with previously assigned literature values [42]. Grafting the polymer resulted in peak shifts not in excess of 0.2°, suggesting no significant changes to the topography of the nanoparticles. Sherrer’s equation was applied to peaks at 31.5°, 56.2°, and 92.0° to determine the approximate sizes of the diffraction planes (FWHM was determined using the Rigaku SmartLab Studio II software suite). The relative standard deviation ranged between 3.5 and 5.8% from the commercial and polymer-grafted nanoparticles, confirming no significant influence of the pre-functionalization with initiators or the polymerization process on the crystallinity of the nanoparticle. One significant change from commercial BaTiO_3_ for the polymer-grafted nanoparticles is the appearance of a broad peak with a 2θ max ranging between 11.6 and 12.1°. This peak is presumed to be due to the presence of the less-ordered polymer on the surface of the nanoparticle.

### Effect of Reaction Times on ATRP-Synthesized Polymer-Grafted Nanoparticles

We evaluated the composition of PS- and PMMA-grafted BaTiO_3_ nanoparticles when the synthesis was performed with the same feed ratio of the monomer to BaTiO_3_-BIB nanoparticles while the ATRP reaction was conducted for 12 h and 24 h. The FT-IR spectra confirmed the observation of characteristic PS and PMMA peaks on the surface of the BaTiO_3_ nanoparticle. Char residue from TGA was used to determine the amount of polymer grafted on the nanoparticles. Table 1 summarizes the characteristics of polymer-grafted nanoparticles synthesized using various methods. The char yield (which is primarily a measure of the inorganic BaTiO_3_) for PS-grafted nanoparticles ranged from 72% to 85%, while the PMMA-grafted nanoparticle char yield ranged from 68.8% to 89%. For both systems, the char yield was found to increase significantly as the polymerization time was reduced from 24 h to 12 h. Using the mass of the residual nanoparticles at 500 °C as a reference, a quantitative analysis of TGA data indicates that the amount of grafted polymer is proportionate to the molecular weight of the grafted polymer. Furthermore, the narrow dispersity obtained for all the grafted nanoparticles (PS- and PMMA-grafted nanoparticles) suggests that the ATRP is an effective controlled polymerization procedure [43]. 

Although the weight loss of PMMA and PS grafted on the nanoparticles is similar based on TGA measurements, a significant difference in the average molecular weight of the two polymers was observed. Figure 7 shows a representative GPC chromatogram of 24 h and 12 h PS and PMMA polymer cleaved from polymer-grafted BaTiO_3_ nanoparticles. PMMA has an average molecular weight of 115,800 g/mol and 230,000 g/mol for 12 h and 24 h reaction times, respectively (Table 1). On the other hand, the average molecular weight of PS was 31,100 g/mol and 83,900 g/mol for 12 h and 24 h reactions, respectively. The molecular weight and char residue data were substituted in Equation (1) and used for calculating graft density. The graft density for PS was found to be 0.12 chains/nm^2^, which was higher than that for PMMA 0.039 and 0.015 chains/nm^2^ for identical polymerization times.

Not all of the immobilized initiators can propagate with monomers to become polymer chains. For those that do, several factors influence the graft density of the polymer on the surface of the nanoparticles, such as the initiator bound on the surface, catalyst concentration, monomer structure (reactivity and stability of the intermediate radical species), solvent, catalyst-to-deactivator ratio used for polymerization, temperature, and time [44]. As reported earlier, the same amounts of initiator-grafted BaTiO_3_ nanoparticles, catalyst, and solvent were used in the synthesis of PS- and PMMA-g-BaTiO_3_ nanoparticles. The high graft density of PS compared to PMMA could be attributed to the polymerization condition. Polymerization of styrene on the nanoparticles was carried out at a higher temperature, which could also alleviate the monomer solubility and, hence, higher graft density. Liu et al. [45] used molecular dynamics simulations to investigate the effect of initiator density and polymerization rate on grafting density. For an ideal living surface-initiated polymerization system, they observed that a faster polymerization rate yields a lower grafting density [45]. Therefore, the lower graft density of PMMA-g-BaTiO_3_ may be due to the higher rate of polymerization of the MMA monomer on BaTiO_3_ compared to the styrene monomer. It is worth noting that this trend is observed for all the reaction procedures adopted in this study. The dispersity of PS-g-BaTiO_3_ and PMMA-g-BaTiO_3_ is (1.609 and 1.514) and (1.504 and 1.568) for the 12 h and 24 h polymerization times, respectively This shows that, even though the reaction time has a major effect on the rate of monomer conversion, which directly influences the molecular weight of the polymer grafted on the nanoparticles, the polydispersity was largely unaffected.

In l ATRP, molecular weight is controlled through the persistent radical effect, which suppresses termination reaction. Initially, a few percent of radical coupling is used to build up a concentration of deactivator, i.e., copper(II) species, which, in sufficient concentrations, will deactivate intermediate radicals faster than the rate at which the radicals can react in termination events [46]. However, chain termination is not completely eliminated during the course of polymerization. Figure 8 summarizes the four possible termination modes that could take place in graft polymerization from the surfaces of discrete particles and in the presence of free chains in solution. There could be termination between free chains in solution, between a free chain and a surface-bound chain, between chains on the same particle surface, and between chains on different particle surfaces. Because some components can diffuse freely in the former two modes, the rates of termination involving free chains (in solution and to the surface) should be faster than those involving only surface-bound chains (intraparticle and interparticle) [30].

Styrene can undergo thermal self-initiation at high temperatures. Because the styrene ATRP polymerization reaction is carried out at a temperature where thermal polymerization is significant (90 °C), some free polystyrene chains could be forming in the solution. This allows termination to occur predominately in solution, though presumably some fraction occurs in the surface-bound polymers. Thus, the surface-bound PS can grow relatively unhindered, and the radical concentration remains constant [30].

On the other hand, MMA does not undergo thermal self-initiation. Because of this, there are little to no free chains in solution during the MMA ATRP. Therefore, most chain termination occurs intraparticle or interparticle. Intra- and interparticle termination could result in the formation of a cross-linked structure over long polymerization. This could result in significantly high molar masses, as observed for PMMA-g-BaTiO_3_ for the 24 h and 12 h polymerization time. Previous observations indicated that we could incorporate some molecular weight control in MMA ATRP by introducing minimal free chains into the solution during the polymerization process.

Two approaches were pursued to introduce free chains in the solution, i.e., ARGET with a sacrificial initiator, as well as ATRP with a sacrificial initiator. Ethyl 2-bromo-2-methylpropionate was used as the sacrificial initiator for the polymerization of styrene and the MMA monomer. This would allow for the growth of some free chains in the solution during the polymerization reaction. The polymerization reactions were conducted for 12 h because 12 h ATRP gave a moderately high molecular weight and better PDI for both PS and PMMA. As presented in Table 1, the TGA residual weight of 12 h ATRP with a sacrificial initiator for PS-g-BaTiO_3_ decreased to 7.2% compared to 10.6% for the 12 h ATRP without sacrificial initiator. Additionally, a drop in the graft density from 0.12 chain/nm^2^ to 0.081 chain/nm^2^ occurred. The drop in the polymer grafting density can be due to the steric hindrance presented by the slightly higher molecular weight in the case of ATRP with a sacrificial initiator.

In the case of PMMA-g-BaTiO_3_ nanoparticles, we observed a more than 50% decrease in the Mw upon the addition of a sacrificial initiator from 115,800 g/mol for normal ATRP to 52,080 g/mol. The grafting density of PMMA-g-BaTiO_3_ nanoparticles increases (0.015 chain/nm^2^ to 0.071 chain/nm^2^) upon the addition of a sacrificial initiator, which suggests that many shorter chains are grafted on the surface of the nanoparticle upon the addition of the sacrificial initiator, as compared to the longer chains we observed in the normal ATRP of PMMA. For both normal ATRP and ATRP with a sacrificial initiator, we observed that the PDI was in a range of 1.5 to 1.7, which shows some good control of the dispersity using this method. However, better control of the polydispersity was obtained using ARGET with a sacrificial initiator, as discussed below.

For the ARGET ATRP with a sacrificial initiator, the Mw of PS-grafted chains increased from 30,000 g/mol to 37,000 g/mol, while the dispersity decreased from 1.5 to 1.259 compared to ATRP with a sacrificial initiator. Similar results were observed for PMMA-g-BaTiO_3_ nanoparticles. For PMMA, the Mw slightly decreased from 52,080 g/mol to 44,620 g/mol, with a drastic drop in the PDI from 1.7 to 1.263 while performing ARGET ATRP with a sacrificial initiator compared to ATRP with a sacrificial initiator. The higher values of Mw indicate that longer PS and PMMA chains are grafted during the ARGET ATRP reaction, and the drop in the dispersity shows good control of the polymerization using ARGET with a sacrificial initiator. It is worth noting that the graft density of both PS- and PMMA-grafted BaTiO_3_ also decreases in ARGET ATRP with a sacrificial initiator compared to ATRP with a sacrificial initiator. For example, the grafting density decreased from 0.081 chain/nm^2^ to 0.067 chain/nm^2^ and from 0.071 chains/nm^2^ to 0.059 chains/nm^2^ for PS- and PMMA-grafted BaTiO_3,_ respectively, indicating that ARGET ATRP favored the grafting of longer polymer chains with lower grafting densities compared with the ATRP method. The low graft densities can be attributed to the fact that surface initiation of BaTiO_3_-BIB in ARGET ATRP is much lower than in ATRP, which is primarily a result of the use of a lower amount (ppm level) of catalyst [47].

## 3. Materials and Methods

BaTiO_3_ nanoparticle (50 nm), 30% aq. H_2_O_2_ solution, (3-aminopropyl)trimethoxysilane (APTMS), α-bromoisobutyryl bromide (BIB), 98%, Sigma Aldrich, St. Louis, MO, USA, ethyl 2-bromo-2-methylpropionate (EBIB, 99%, Sigma Aldrich), anhydrous N,N’-dimethylformamide (DMF), N,N,N′,N′′,N′′-pentamethyldiethylenetriamine (PMDETA), stannous octoate, copper(II) bromide, copper(I) bromide, 40% hydrofluoric acid (HF), toluene, tetrahydrofuran (THF), an dichloromethane (DCM) were purchased from Sigma-Aldrich Chemical Company. Styrene and methylmethacrylate (MMA) were received from Thomas Scientific and distilled under a vacuum to remove the inhibitor before use, whereas all other reagents and solvents were used as received. 

### 3.1. Synthesis of Core–Shell Nanoparticles

#### 3.1.1. Surface-Initiated Atomic Transfer Radical Polymerization (ATRP)

For the synthesis of homopolymer core–shell nanoparticles using ATRP procedure, we slightly modified the *grafting-from* approach reported in the literature [35]. In our procedure, we used (3-aminopropyl) trimethoxysilane (APTMS) as the silane coupling agent instead of γ-aminopropyl triethoxysilane, as reported in [35], and the monomers were distilled under vacuum before use without washing with NaOH solution. Additionally, we performed sonication and stirred the mixture for 1 h prior to the reaction. Further, our polymer-grafted nanoparticles were dried under vacuum for 48 h in ambient conditions. Details about adopting the modified ATRP procedure for the synthesis of PMMA-g-BaTiO_3_ have already been reported elsewhere [35,38]. 

Briefly, the nanoparticle was anchored with ATRP initiator by performing H_2_O_2_ hydroxylation, APTMS silylation followed by treatment with bromoisobutyryl bromide. The ATRP initiator-functionalized nanoparticle was then used for surface-initiated atom transfer polymerization (SI-ATRP) of monomer with various reaction times to obtain polymer-grafted nanoparticles with varying molecular weights and graft densities. To graft polystyrene (PS) on the surface of 50 nm BaTiO_3_ nanoparticle, initiator-anchored BaTiO_3_-BIB and 12 mL DMF were placed in a Schlenk flask and interchangeably stirred and sonicated for 1 h. CuBr was added and the flask was sealed with a rubber plug. The oxygen in the flask was removed by backfilling with N_2_ using a balloon. Styrene and (0.06 g, 72 µL) PMDETA were added via syringe to the reaction flask, and a freeze–thaw pump cycle was performed three times to degas the reaction mixture. The ratio of styrene:BaTiO_3_-BIB:CuBr was kept as 13:1:0.5 [35]. This reaction mixture was then stirred at 90 °C for 24 h. After 24 h, the reaction was stopped by placing the flask in liquid nitrogen and exposing the contents to air. The obtained PS-grafted BaTiO_3_ (PS-g-BaTiO_3_) nanoparticles after polymerization and centrifugation were re-dispersed in THF and washed several times in acetone to remove the un-grafted polymer and unreacted monomer from the PS-g-BaTiO_3_ nanoparticles. Both the polymer-grafted nanoparticles and the cleaved polymer from the grafted nanoparticles were subjected to extensive characterization. Additionally, we studied the effect of reaction time on synthesizing PS and PMMA-grafted BaTiO_3_ nanoparticles of varying shell thickness and shell coverage by conducting polymerization for 12 and 24 h, while all other parameters were kept constant. We also adopted a minor modification to the procedure of ATRP technique for synthesis of PS-g-BaTiO_3_ and PMMA-g-BaTiO_3,_ respectively, by adding ethyl 2-bromo-2-methylpropionate (EBIB) to the reaction mixture prior to degassing the system via freeze–pump–thaw cycles and purging with nitrogen gas so as to study the effect of sacrificial initiator on the structure of polymer grafts tethered to nanoparticles. 

#### 3.1.2. Activator Regeneration via Electron Transfer (ARGET) ATRP with Sacrificial Initiator for Synthesis of PS- and PMMA-Grafted BaTiO_3_ Nanoparticles 

Polymerization of PMMA-g-BaTiO_3_ nanoparticles via ARGET ATRP was conducted by using a copper(II) complex that was reduced in situ using stannous octoate as the reducing agent. Initiator-anchored BaTiO_3_-BIB and 12 mL DMF were placed in a Schlenk flask, and the flask was sealed with a rubber plug. The reaction mixture was sonicated for 45 min using an ultrasonicator, after which CuBr_2_ and MMA monomer were added, and the mixture was stirred for 15 min. The ratio of monomer/initiator-modified nanoparticles/CuBr_2_ was kept as 13:1.0:0.5 [25]. To the flask, 470µL PMDETA was added via syringe alongside 5 µL ethyl 2-bromo-2-methylpropionate. After the system was degassed through two freeze–pump–thaw cycles and purged with nitrogen gas, stannous octoate (72 µL) was added via syringe under N_2_ atmosphere to trigger the polymerization prior to performing a third freeze–pump–thaw cycle. The Schlenk flask was then placed in an oil bath set at 65 °C for 12 h. The reaction was quenched by immersing it into liquid nitrogen and exposing the contents to air. The product was recovered via centrifugation at 10,000 rpm for 10 min. The recovered polymer-grafted BaTiO_3_ nanoparticles were redispersed in acetone to remove un-grafted PMMA and unreacted monomer. 

The synthesis of PS-g-BaTiO_3_ nanoparticle was similar to that of PMMA-g-BaTiO_3_ nanoparticle using ARGET ATRP method except that the polymerization was carried out at 90 °C for 12 h in an oil bath. The workup to recover PS-g-BaTiO_3_ nanoparticles was identical to the procedure described earlier. 

The washed and recovered grafted nanoparticles were dried under a vacuum for 48 h. The product was used for FT-IR, TGA, TEM, and PXRD characterization, in addition to cleavage experiments. 

### 3.2. Characterization of Polymer-Grafted BaTiO_3_ Nanoparticles

In order to characterize the grafted polymer on the surface of the nanoparticles, the polymer-grafted nanoparticles were subjected to a cleavage experiment using HF in THF. Details about cleaving the polymer from the grafted nanoparticles and recovery of PMMA or PS via precipitation in hexane can be found elsewhere [38]. The recovered cleaved PS and PMMA were characterized using nuclear magnetic resonance (NMR) and gel permeation chromatography (GPC) techniques. 

Molecular weight measurements of cleaved polymers were performed using gel permeation chromatography (GPC) on Agilent Technologies 1260 Infinity. The samples obtained from the cleavage experiment were dissolved in THF and filtered using a 0.2 μm syringe filter. Polystyrene standards were used for calibration. THF was used as the eluent, and the cleaved polymer concentration was maintained at about 1 mg mL^−1^. 

^1^H NMR spectra of cleaved homopolymer were recorded on Bruker AVANCE 400 MHz NMR spectrometer using chloroform-d as the solvent. Chemical shifts given in parts per million (δ ppm) were relative to tetramethylsilane (TMS) and referenced to residual CDCl_3_ at 7.26 ppm. 

Fourier-transform infrared spectroscopy (FT-IR) spectra of all the precursors used for polymer-grafted nanoparticles (BaTiO_3_-OH, BaTiO_3_-APTMS, and BaTiO_3_-BIB) as well as the polymer-grafted nanoparticle were collected using a Perkin-Elmer ATR-FT-IR Spectrum 100 Series. A minimum of 16 scans were collected in a range of 4000 to 650 cm^−1^, and triplicate measurements were performed to confirm the reproducibility of IR spectra.

Thermogravimetric analysis (TGA) of all the precursors and polymer-grafted nanoparticles was conducted using a TGA/DTA 320 Seiko instrument under a nitrogen atmosphere (flow rate of 100 mL min^−1^). The thermogram was generated by placing nearly 10 mg of samples in a pan and heating it from 30 °C to 600 °C at a heating rate of 10 °C per minute. The char yields were used to calculate the amount of polymer grafted on the nanoparticle as well as the graft density of the core–shell structure. The graft density was calculated using Equation (1) below.
(1)$$\begin{array}{cc} \sigma_{TGA}= & \frac{\frac{wt\;{\%}_{shell}\;}{wt\;\%{\;}_{core}}\;\rho_{core}\frac{4\;}{3}\pi r_{core}{}^{3}N_{A}}{M_{w}4\pi r_{core}{}^{2}} \end{array}$$
where mass % of shell = difference in char yield of BT nanoparticle and that of PGNPs; mass % of core = Char yield of BT nanoparticle, *N_A_* = Avogadro’s number, *ρ_core_* = 6.09 g/cm^3^ which represent the density of bulk BaTiO_3_, *r_core_* = 50 nm, corresponding to the radius of BaTiO_3_ assuming a spherical shape, and *M_W_* = Mass average molecular mass of cleaved polymer chains.

Transmission electron microscopy (TEM) analysis was conducted to characterize the shell thickness and dispersion of the homopolymer-grafted nanoparticle in film-casting solvent. To prepare TEM samples, about 2 mg of the grafted nanoparticle samples was dispersed in 3 mL of DMF and sonicated for 1 h. The solution was further diluted using serial dilution. The diluted solution was cast on a 200-mesh carbon-coated copper grid (Electron Microscopy Science, Hatfield, PA, USA). The cast film was left overnight in air at room temperature to dry. The dried sample was preserved in a vacuum desiccator until further use. TEM micrographs of dried samples were collected with a JEM 2100 LaB6 TEM at Nanoscale Imaging, Spectroscopy, and Properties Facility at University of Maryland, College Park.

Powder X-ray diffractometry (XRD) was performed to determine what, if any, influence the polymerization process had on the crystal structure of BaTiO_3_ nanoparticles. The wide-angle X-ray diffraction of the functionalized and polymer-grafted BaTiO_3_ nanoparticles was collected using a Rigaku MiniFlex X-ray powder diffractometer (Cu Kα radiation λ = 1.5418 Å) from 3° to 90° at a scan rate of 0.0166 °/s (λ = 1.5418 Å) (40 kV, 15 mA), with a step size of 0.010° and a step time of 0.6 s. 

## 4. Conclusions

In the present study, ARGET ATRP with a sacrificial initiator and normal ATRP with and without a sacrificial initiator were employed for the synthesis of PMMA/PS-grafted BaTiO_3_ nanoparticles. PMMA/PS-grafted BaTiO_3_ nanoparticles were characterized using FT-IR and ^1^H-NMR to confirm their chemical structure, while TGA was used for estimation of the % of polymer grafted on the NP surface. The cleaved polymer samples were analyzed via GPC for the determination of molar masses and dispersity. TEM analysis confirmed the successful grafting of PMMA and PS on the surface of BaTiO_3_ nanoparticles, with shell thickness in a range of 5–17 nm. The effect of time on the molecular weight and dispersity was also investigated, and it revealed livingness of polymerization reactions, and molecular weights increased with polymerization time. The variation in the molecular weight and graft density with respect to polymerization conditions, monomer reactivity, monomer solubility in solvent, and time was explained, and plausible pathways for polymerization termination were provided. It was observed that a sacrificial initiator played an important role in controlling the dispersity and lowering the molecular weights, as compared to ATRP without a sacrificial initiator. Further, ARGET with a sacrificial initiator had better control of polymer grafts of both MMA and styrene on BaTiO_3_ nanoparticles. The present results provide a la carte approaches to synthesize polymer-grafted BaTiO_3_ nanoparticles with defined molecular weights and narrow dispersity. The ability to tune the structure of polymer-grafted nanoparticle hybrids, depending on the selection of the methodology, opens the door for material design, depending upon the intended application and the requirements.

## Data Availability

Not applicable.

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
