# Peer review of "Comparative Study of Polymer-Grafted BaTiO3 Nanoparticles Synthesized Using Normal ATRP as Well as ATRP and ARGET-ATRP with Sacrificial Initiator with a Focus on Controlling the Polymer Graft Density and Molecular Weight"

_molecules, 2023, doi:10.3390/molecules28114444_

Round 1

Reviewer 1 Report

The manuscript reports the synthesis of PMMA grafted- and PS-grafted BaTiO3 nanoparticles by using ATRP, ATRP with sacrificial initiator, and ARGET-ATRP with sacrificial initiator, with a focus on controlling the polymer graft density and molecular weight. The polymerization procedures proved to have much influence on the structures of nanoparticle hybrids. The use of a sacrificial initiator along with ARGET-ATRP was found to provide the best control in achieving lower molecular weight and narrower dispersity for both PS (37,870 g/mol and PDI of 1.259) and PMMA (44,620 g/mol and PDI of 1.263) nanoparticle hybrid systems. The results are interesting and offer some important rules for designing well-defined nanoparticle hybrids. I recommend the manuscript to be published in the journal after the authors address the following issues:

1) It is better to change the title of the manuscript to “Comparative study of polymer-grafted BaTiO3 nanoparticles synthesized using ATRP as well as ATRP and ARGET-ATRP with sacrificial initiator with a focus on controlling the polymer graft density and molecular weight”.

2) A scheme should be presented in the manuscript to schematically show the synthetic procedure of the polymer-grafted BaTiO3 nanoparticles.

3)   In page 5, it seems that mass% of shell should be equivalent to (Char yield of BT nanoparticles) minus (Char yield of the PGNPs) in equation (1).

4)  In Figure 1B, BaTiO3-OH showed much larger weight loss than bare BaTiO3 and other functionalized BaTiO3 in the beginning during the TGA analyses, which might be attributed to its containing some water. This should be mentioned in the manuscript.

5)  Figure 2A and B should be the FTIR spectra of PS and PMMA-grafted BaTiO3 nanoparticles, respectively. The position of Figure 1A and B is reversed.

6)   In Figure 3, which PS-grafted BaTiO3 nanoparticles (In Table 1) are the cleaved PS derived from? This should be mentioned in the legend of the Figure. Similarly, in the legend of Figure 4, the polymer-grafted nanoparticles should also be specified.

7)  In Table 1, the relationship between TGA char yield and polymer weight loss should be mentioned.

8)  In the 5th paragraph in page 12, the molecular weight and dispersity values of the grafted polymers are different from those listed in Table 1.

Author Response

Reviewer 1

Revision requested.

The manuscript reports the synthesis of PMMA grafted- and PS-grafted BaTiO3 nanoparticles by using ATRP, ATRP with sacrificial initiator, and ARGET-ATRP with sacrificial initiator, with a focus on controlling the polymer graft density and molecular weight. The polymerization procedures proved to have much influence on the structures of nanoparticle hybrids. The use of a sacrificial initiator along with ARGET-ATRP was found to provide the best control in achieving lower molecular weight and narrower dispersity for both PS (37,870 g/mol and PDI of 1.259) and PMMA (44,620 g/mol and PDI of 1.263) nanoparticle hybrid systems. The results are interesting and offer some important rules for designing well-defined nanoparticle hybrids. I recommend the manuscript to be published in the journal after the authors address the following issues:

Response: We would like to thank the Reviewer for kind appreciation of the work and critical evaluation of the work. We have revised the manuscript by taking into consideration the helpful comments from the reviewers.

  • It is better to change the title of the manuscript to “Comparative study of polymer-grafted BaTiO3nanoparticles synthesized using ATRP as well as ATRP and ARGET-ATRP with sacrificial initiator with a focus on controlling the polymer graft density and molecular weight”.

Response: We would like to thank Reviewer for the suggestion, the correction in the tittle has been made in the revised manuscript.

  • A scheme should be presented in the manuscript to schematically show the synthetic procedure of the polymer-grafted BaTiO3

Response: The scheme for the synthetic procedure of the polymer-grafted BaTiO3 nanoparticles has been included in the revised manuscript.

  • In page 5, it seems that mass% of shell should be equivalent to (Char yield of BT nanoparticles) minus (Char yield of the PGNPs) in equation (1).

Response: Authors would like to apologize for the error. The correction has been made in the revised manuscript. The mass % of the shell has been defined as “ difference in char yield of BT nanoparticle and that of PGNPs” in the revised manuscript.

  • In Figure 1B, BaTiO3-OH showed much larger weight loss than bare BaTiO3and other functionalized BaTiO3 in the beginning during the TGA analyses, which might be attributed to its containing some water. This should be mentioned in the manuscript.

Response: The correction has been made in the revised manuscript including the statement “BaTiO3-OH showed much larger weight loss than bare BaTiO3 and other functionalized BaTiO3 in the beginning during the TGA analyses, which might be attributed to it containing some water.”

5)  Figure 2A and B should be the FTIR spectra of PS and PMMA-grafted BaTiO3 nanoparticles, respectively. The position of Figure 1A and B is reversed.

Response: We apologize for the mistake in the Figure caption. The correction in the caption has been made in the revised manuscript

6 In Figure 3, which PS-grafted BaTiO3 nanoparticles (In Table 1) are the cleaved PS derived from? This should be mentioned in the legend of the Figure. Similarly, in the legend of Figure 4, the polymer-grafted nanoparticles should also be specified.

Response: The captions for Figure 3 and 4 have been modified as per suggestion.

  • In Table 1, the relationship between TGA char yield and polymer weight loss should be mentioned.

Response: The correction has been made

  • In the 5thparagraph in page 12, the molecular weight and dispersity values of the grafted polymers are different from those listed in Table 1.

Response: The molecular weigh and dispersity data (16,990 g/mol and 1.213) on the page corresponds to free chain obtained from polymerization solution, therefore it has not been included in Table 1. The molecular weight data of polymer chains cleaved from PGNPs is included in Table 1

Reviewer 2 Report

This manuscript presented the “grafting from” approach to grow polymer chains from the surface of BaTiO3 nanoparticles using surface initiated ATRP. Three types of ATRP methods were used: normal ATRP, “ARGET” ATRP and ATRP in the presence of sacrificial initiators. The polymers grafted onto the surface of nanoparticles were given sufficient characterization analysis. This is an interesting work, and the authors presented the results clearly and concisely. Although the reviewer still suggests that authors should provide a bit more detailed explanation to the following questions.

1.     The authors are advised to do more careful proofread, since there are some minor formatting errors. For example, the molecular weight should be presented as “44,620” instead of “44.620” in the abstract. Same for the Table 1 (PS-g-BaTiO3 ARGET 12h, Mn should be 30,090 instead of 30.090. Another example that the authors should pay attention is that “ATRP ARGET” should be “ARGET ATRP” (page 2), since “ARGET” essentially describes what type of ATRP there is.

2.     The title might be a little misleading, since ARGET is also a sub-category of ATRP. So, I would suggest using “normal ATRP, ARGET ATRP, and ATRP with sacrificial initiator”.

3.     The reviewer suggests that the authors should explain the importance of BaTiO3 and why is there a need to graft from the surface of BaTiO3. The authors did a good job explaining polymer-nanoparticle composite materials in general, but the reviewer thinks it is necessary to explain the potential applications of BaTiO3 nanoparticles specifically, in order to more clearly identify the importance of this work. This could be added to the introduction section.

4.     In the experimental section (page 3), the polymerization condition stated that the ratio of styrene: BaTiO3-BiB is 13:1. This doesn’t seem to be reasonable when compared with the molecular weight information. The styrene ratio should be much higher. Also, it is better to clarify that this is the molar ratio.

5.     Page 6: The discussion stated Fig 2A is PMMA and Fig 2B is PS, but in fact Fig 2A is PS and Fig 2B is PMMA-grafted polymers.

6.     Another comment on the FTIR spectra of PS in page 6: the peak at ~700 cm-1 is also usually associated with aromatic structures, especially with styrene. This would be a better indication of more styrene in the 24h sample, since the unreacted styrene monomer also has C=C at ~ 1490 cm-1 which might interfere with the interpretation of the spectra.

7.     The reviewer noticed that no GPC chromatogram or molecular weight distribution graph were shown. This should be an essential piece of information since it will more clearly show the evolution of MWD during polymerization, as well as demonstrate whether there is any skewness in the MWD.

8.     This is not necessary but would be nice to have: any TEM images of the unfunctionalized BaTiO3 nanoparticles to compare with PMMA/PS-g-BaTiO3? Are there any aggregations of particles in the pristine material?

9.     TEM: is there a reason why PMMA-g-BaTiO3 and PS-g-BaTiO3 showed similar contrast? Usually, styrene-based polymers would give a higher contrast in TEM images. Did the authors use any staining reagent to improve the contrast?

10.  Page 10: the authors mentioned the MW differences between PMMA and PS. However, the reviewer believes that an important factor here is that only “apparent” molecular weights based on a linear calibration of narrow polystyrene standards were measured. Since PMMA and PS with the same molecular weight would still exhibit different hydrodynamic volume in the eluent (THF), the difference in apparent molecular weight does not definitively indicate a difference in the absolute molecular weight. The reviewer wonders if a PMMA calibration would be available for calculating the MW of PMMA (Agilent should have a bunch of different PMMA calibration standards), or if MALS detector is available for calculating the absolute molecular weight? If not, the authors should at least mention that the apparent molecular weight does not indicate the “real” molecular weight differences.

11.  Page 11: “the lower graft density of PMMA-g-BaTiO3 may be due to higher rate of polymerization of MMA monomer on BaTiO3 compared to styrene monomer”. Is there a reference for this polymerization rate differences?

12.  Page 12: the authors mentioned that “there are little to no free chains in solution during MMA ATRP”. Is there any experimental evidence for this?

Dear editor,

Please refer to the comment to authors.

Thanks!

Author Response

Reviewer 2

Revision requested

This manuscript presented the “grafting from” approach to grow polymer chains from the surface of BaTiO3 nanoparticles using surface initiated ATRP. Three types of ATRP methods were used: normal ATRP, “ARGET” ATRP and ATRP in the presence of sacrificial initiators. The polymers grafted onto the surface of nanoparticles were given sufficient characterization analysis. This is an interesting work, and the authors presented the results clearly and concisely. Although the reviewer still suggests that authors should provide a bit more detailed explanation to the following questions.

Response: Authors would like to thank the Reviewer for kind appreciation of the work and favorable recommendation. We have revised the manuscript by taking into consideration the helpful comments from the reviewers.

  1. The authors are advised to do more careful proofread, since there are some minor formatting errors. For example, the molecular weight should be presented as “44,620” instead of “44.620” in the abstract. Same for the Table 1 (PS-g-BaTiO3 ARGET 12h, Mn should be 30,090 instead of 30.090. Another example that the authors should pay attention is that “ATRP ARGET” should be “ARGET ATRP” (page 2), since “ARGET” essentially describes what type of ATRP there is.

Response: The correction has been made in the revised manuscript.

  1. The title might be a little misleading, since ARGET is also a sub-category of ATRP. So, I would suggest using “normal ATRP, ARGET ATRP, and ATRP with sacrificial initiator”.

Response: The correction has been made in the revised manuscript.

  1. The reviewer suggests that the authors should explain the importance of BaTiO3 and why is there a need to graft from the surface of BaTiO3. The authors did a good job explaining polymer-nanoparticle composite materials in general, but the reviewer thinks it is necessary to explain the potential applications of BaTiO3 nanoparticles specifically, in order to more clearly identify the importance of this work. This could be added to the introduction section.

Response: The statement for importance BaTiO3 were included in introduction “BaTiO3 nanoparticles are of particular interest among dielectric materials due to their superior dielectric, piezoelectric, and ferroelectric properties, flexibility in adjusting their shape, and widespread use in energy storage, electronics, and device applications. As a result, numerous methodologies for creating BaTiO3-based nanocomposites have been devised in order to increase the compatibility of the nanoparticles with the polymer matrix, the spatial dispersion of the nanoparticles in the matrix, and the permittivity contrast between polymer and nanoparticles.

  1. In the experimental section (page 3), the polymerization condition stated that the ratio of styrene: BaTiO3-BiB is 13:1. This doesn’t seem to be reasonable when compared with the molecular weight information. The styrene ratio should be much higher. Also, it is better to clarify that this is the molar ratio.

Response: The ratio of “styrene: BaTiO3-BiB is 13:1” mentioned in the manuscript corresponds to the weight ratio of Styrene and BIB modified BT nanoparticles. A very small weight fraction (less than 1%) of BIB (initiator functionality) was attached on the BT nanoparticle surface. Therefore, the ratio styrene: BaTiO3-BiB is 13:1 seems to be high.

  1. Page 6: The discussion stated Fig 2A is PMMA and Fig 2B is PS, but in fact Fig 2A is PS and Fig 2B is PMMA-grafted polymers.

Response: The correction has been made in the revised manuscript.

  1. Another comment on the FTIR spectra of PS in page 6: the peak at ~700 cm-1 is also usually associated with aromatic structures, especially with styrene. This would be a better indication of more styrene in the 24h sample, since the unreacted styrene monomer also has C=C at ~ 1490 cm-1 which might interfere with the interpretation of the spectra.

Response: We have purified PGNPs with several times washings and dried under vacuum to make sure that there is no presence of unreacted styrene. This is supported by the lack of the styrene alkene stretch at 1630 cm-1. Therefore, the peak at 1490 cm-1 could be attributed to C=C aromatic. As per reviewers’ suggestions, we have also mentioned increment in the peak at 751 cm-1 could be due to the more grafting of PS onto BaTiO3 nanoparticles.

  1. The reviewer noticed that no GPC chromatogram or molecular weight distribution graph were shown. This should be an essential piece of information since it will more clearly show the evolution of MWD during polymerization, as well as demonstrate whether there is any skewness in the MWD.

Response: GPC chromatogram or molecular weight distribution graph have been included.

  1. This is not necessary but would be nice to have: any TEM images of the unfunctionalized BaTiO3 nanoparticles to compare with PMMA/PS-g-BaTiO3? Are there any aggregations of particles in the pristine material?

Response: No effort was made to study to compare the unfunctionalized BaTiO3 nanoparticles to compare with PMMA/PS-g-BaTiO3.

  1. TEM: is there a reason why PMMA-g-BaTiO3 and PS-g-BaTiO3 showed similar contrast? Usually, styrene-based polymers would give a higher contrast in TEM images. Did the authors use any staining reagent to improve the contrast?

Response: No staining agents were used for TEM analysis. However, we did not observe large variation in the contrast for PMMA-g-BaTiO3 and PS-g-BaTiO3.

  1. Page 10: the authors mentioned the MW differences between PMMA and PS. However, the reviewer believes that an important factor here is that only “apparent” molecular weights based on a linear calibration of narrow polystyrene standards were measured. Since PMMA and PS with the same molecular weight would still exhibit different hydrodynamic volume in the eluent (THF), the difference in apparent molecular weight does not definitively indicate a difference in the absolute molecular weight. The reviewer wonders if a PMMA calibration would be available for calculating the MW of PMMA (Agilent should have a bunch of different PMMA calibration standards), or if MALS detector is available for calculating the absolute molecular weight? If not, the authors should at least mention that the apparent molecular weight does not indicate the “real” molecular weight differences.

Response: The GPC molecular weights were measured with relative polystyrene standards.

11.  Page 11: “the lower graft density of PMMA-g-BaTiO3 may be due to higher rate of polymerization of MMA monomer on BaTiO3 compared to styrene monomer”. Is there a reference for this polymerization rate differences?

Response: It has been reported that in case of surface initiated ARGET polymerization of MMA and Styrene from the surface of BaTiO3 NPs, MMA is reactive monomer than styrene leading to more amount of polymer grafting in short time. (ACS Appl. Mater. Interfaces 2014, 6, 5, 3477–3482).

  1. Page 12: the authors mentioned that “there are little to no free chains in solution during MMA ATRP”. Is there any experimental evidence for this?

Response: For both experiment that was done with and without sacrificial initiator, the supernatant from this experiment were concentrated after centrifugation and subjected to precipitation using cold hexane. We observed precipitation for the experiment with sacrificial initiator similar to cleaved polymer that was precipitated in cold hexane. However, we did not observe any precipitation for MMA polymerization experiment without sacrificial initiator.